# Analogous Diamondene Nanotube Structure Prediction Based on Molecular Dynamics and First-Principle Calculations

**DOI:** 10.3390/nano10050846

**Published:** 2020-04-28

**Authors:** Xin Zhou, Haifang Cai, Chunwei Hu, Jiao Shi, Zongli Li, Kun Cai

**Affiliations:** 1College of Water Resources and Architectural Engineering, Northwest A&F University, Yangling 712100, China; 2State Key Laboratory of Mechanics and Control of Mechanical Structures and MOE Key Laboratory for Intelligent Nano Materials and Devices, College of Aerospace Engineering, Nanjing University of Aeronautics and Astronautics, Nanjing 210016, China; 3State Key Laboratory of Structural Analysis for Industrial Equipment, Dalian University of Technology, Dalian 116024, China; 4School of Engineering, RMIT University, Melbourne VIC 3001, Australia

**Keywords:** concentric twin nanotube, diamondene nanotube, buckling, molecular dynamics, first principles

## Abstract

A concentric twin tube (CTT) can be built by placing a carbon nanotube (CNT) in another identical CNT. Different from diamondene nanotubes, a stable CTT has no inter-shell covalent bond. As a prestressed double-walled nanotube, CTT has a lower structural stability at a finite temperature. According to the molecular dynamics and first-principle calculations, (a) CTTs have three types of relaxed configurations. In a type III CTT, the inner tube buckles to produce a V-shaped cross-section, and the outer tube may be convex or concave. (b) The minimal radii of relaxed zigzag and armchair CTTs with concave outer tubes were found. (c) After relaxation, the circumferences and areas of the two tubes in a type III CTT are different from those of the corresponding ideal CNT. The area change rate (A-CR) and circumference change rate (C-CR) of the outer tube are the first-order Gaussian function of the radius of the ideal CNT (which forms the CTT), and tends to be 73.3% of A-CR or 95.3% of C-CR, respectively. For the inner tube of a CTT, the A-CR is between 29.3% and 37.0%, and the C-CR is close to 95.8%. (d) The temperature slightly influences the findings given above.

## 1. Introduction

Low-dimensional carbon materials are classified into several groups, e.g., fullerene [1,2] as zero-dimensional material; a carbon nanotube (CNT) [3] as one-dimensional material; and two-dimensional materials containing graphene [4,5], graphane [6,7,8], diamondene [9,10,11]/diamane (with hydrogen atoms) [12,13,14], and diamond films [13,15,16,17,18,19]. Benefitting from the electron configurations, carbon materials have excellent properties in physics. For example, graphene has perfect thermal and electrical conductivities [20,21,22,23], and an extremely high strength and modulus [24,25,26]. Diamond has a high hardness and optical properties. Graphane and diamondene are different from both graphene (sp^2^ material) and diamond (sp^3^ material), because of their composited sp^2^ and sp^3^ bonds, respectively. Hence, their properties are also different. For instance, graphane exhibits anisotropic properties [27], relatively lower mechanical properties [28], a high sensitivity to temperature [29], and an adjustable band gap [30]. Diamondene can be fabricated by compressing two or more layers of graphene [17,19,31,32], which have been approved by detecting the sp2-to-sp3 phase transition via the Raman spectrum [9,17]. It has a high strength [9], hardness [9,18], and pressure-controlled reversible electrical conductivity [32]. Chernozatonskii et al. [12,15] found that the stability of diamane/diamondene lies between that of graphene and graphane. However, diamondene is unstable at ambient temperature once the hydrogen atoms are removed [10], and the interlayer C-C bonds collapse as a result.

By imitating the method for fabricating a CNT from a graphene ribbon, graphane and diamondene can also be predicted to be used to form nanotubes. For instance, Wen et al. [33] found that the armchair nanotube from graphane has a higher stability than the zigzag nanotube, and more C-H bonds on a single-walled graphane tube will enhance the structural stability. Cai and colleagues [34,35,36] predicted the thermal and mechanical properties of diamondene nanotubes (DNTs). The critical stable temperature of DNTs is higher than that of diamondene itself, but interlayer sp3-sp3 bonds are brittle. The new nanotubes may have different physical properties to those of CNTs, e.g., DNTs display a softening-hardening process under uniaxial tension and are more stable than similar double-walled CNTs under pressure. Therefore, these nanotubes could be a potential backing material and flexible material in the design of nanodevices based on their unique properties.

In the structural optimization of a DNT via first-principle calculations, the nanotube experiences a series of transition states. After optimization, it may become a prestressed double-walled nanotube [37,38,39,40]. Based on this phenomenon, a new double-walled tube is proposed in the present study. The tube is formed by two identical CNTs, and called a concentric twin nanotube (CTT). The structural stability of CTTs is discussed by considering the effects of the radii of the corresponding CNTs, chiral indexes, and temperature. The configurations of the relaxed CTTs are also evaluated by using the circumference change rate (C-CR) and area change rate (A-CR).

## 2. Models and Methods

### 2.1. Models

If a diamondene nanotube (Figure 1b,c) is considered as a double-walled tube, its inner and outer shells will be covalently bonded. The interlayer covalent bond in a DNT is brittle [36], and thus, the sp^2^-sp^3^ composite carbon material has a lower thermal stability [10]. By uniformly expanding the outer tube (black atoms in Figure 1b,c) and shrinking the inner tube of a DNT (red atoms in Figure 1b,c) along the radial direction to keep the inter-shell distance at a value of 3.4 Å, a CTT can be obtained. The CTT structures are shown in Figure 1d,e. One can also obtain a CTT from two identical CNTs by simultaneous expansion and shrinkage of the two CNTs. Therefore, a CTT has no inter-shell bonds, which is different from both DNTs and common double-walled carbon nanotubes due to the same chiral index of the two tubes. To clarify the notations of the tubes, the chiral indexes of the CTTs and DNTs are expressed by double square brackets and double round brackets, respectively, e.g., [[a,a]] refers to armchair CTT [[a,a]], and ((a,a)) represents armchair DNT ((a,a)).

### 2.2. Methodology

The stable configurations of CTTs were investigated by both the molecular dynamics (MD) method and first-principle calculations in this study. MD simulations were performed on open source code LAMMPS [41]. The models were put in a periodic box with a square cross-section and the side length of 70 Å, which was higher than the maximal diameter of the tubes. The adaptive intermolecular reactive empirical bond order potential(AIREBO)-Morse empirical potential was chosen to describe the atomic interaction [42]. Morse potential is employed to describe non-bonding interactions at an extremely high pressure.
(1)$$U_{ij}(r)=-\varepsilon_{ij}\left[1-{\left(1-e^{-\alpha_{ij}\left(r-r_{ij}^{eq}\right)}\right)}^{2}\right],$$
where the *ij* indices indicate the *i*th and *j*th atoms, *ε* and *r^eq^* define the depth and location of the minimum energy, and parameter *α* modifies the curvature of the potential energy at its minimum separation. In simulations, a Nosé–Hoover thermostat [43,44] was used to control the temperature of the system in an NVT ensemble (N: Number of atoms; V: Volume of system; T: Temperature). The timestep was set to 0.1 fs. Each simulation contained 5,000,000 timesteps, and the results were recorded every 1000 steps (0.1 ps). Temperature effects will be discussed by considering four values, i.e., 8 K (ultralow temperature), 100 K (low temperature), 300 K (ambient temperature), and 500 K (high temperature).

The first-principle calculations were conducted on code VASP. In our models of CTTs, the outer and inner tube had the same number of atoms, i.e., 4 × n for an A-n tube or 4 × m for a Z-m tube in a cuboid unit cell (*a* × *b* × *c*), and the values of *a* and *b* ensured that the vacuum thickness along the radial direction was higher than 10 Å, to avoid interaction with neighboring periodic cells. The value of *c* was 2.511 Å for A- CTTs and 4.350 Å for Z- CTTs. All first-principle computations were performed within the framework of density-functional theory (DFT) using the projector augmented wave method with the Perdew–Burke–Ernzerhof (PBE) exchange-correlation functional [45,46,47] while the plane wave energy cut-off was set to 550 eV. The influence of van der Waals (vdW) interactions was considered by using a modified version of vdW-DF, referred to as “optB86b-vdW”. The PBE exchange functional of the original vdW-DF of Dion et al., was replaced with the optB86b exchange functional, in order to yield more accurate equilibrium interatomic distances and energies for a wide range of systems [48,49]. The whole system was relaxed by using a conjugate-gradient algorithm until the variation of the system energy was less than 1.0 × 10^−4^ eV between two adjacent ionic steps. For all relaxation processes, special k-points were sampled on 1 × 1 × 6 Gamma centered Monkhorst-Pack grids.

The initial models of the four types of nanotubes presented in Figure 1 were obtained by a geometric mapping approach [50]. The axial length of the initial CTT models for MD simulation was five times the length of the unit cell which was selected for the first-principle calculations when considering the expensive computational cost. In an MD simulation, a carbon atom on the outer tube is fixed to control the position of CTTs. For simplicity, in this work, “A-n” represents A-CTT [[n,n]] and “Z-m” represents Z-CTT [[m,0]]. The subscripts “out” and “in” represent the outer tube and inner tube, respectively. A CTT could be fabricated via the assembly of two identical CNTs with electron radiation [51,52,53].

### 2.3. Parameter Definition

For a system, its potential energy (PE) can be used to characterize its stability. Compared with the total potential energy, the potential energy per atom (PEA) defined below is suitable for a comparison of the states of systems with different numbers of atoms.
(2)PEA=PE/N,
where *N* is the number of atoms in the system.

Changing the configuration of a system can be quantitatively described by the variation of PE, i.e., the difference between the current and the initial PE of the system. Similarly, the states of different systems can be compared by using the variation of potential energy per atom (VPEA):(3)$$\mathrm{VPEA}=\mathrm{PEA}(t)-\mathrm{PEA}(t_{0}),$$
where *t* and *t*_0_ are the current and initial moments in the simulation, respectively.

To evaluate the variation of a system’s configuration, *β*, as the circumference change rate (C-CR), and *γ*, as the area change rate (A-CR), can be defined as
(4)$$\begin{array}{l} \beta =\left(C/C_{0}\right)\times 100\%,\\ \gamma =\left(A/A_{0}\right)\times 100\%. \end{array}$$
where *A* and *C* are the area and circumference of the deformed tube with an initial area of *A*_0_ and initial circumference of *C*_0_.

## 3. Results and Discussion

### 3.1. Stable Configurations of CTTs at 8 K

By relaxing the CTTs with initial configurations, as shown in Figure 1d,e, the final configurations of these tubes were obtained by MD simulations. Among the final configurations of the Z-CTTs from [[10,0]] (i.e., Z-10 in Figure 2) to [[63,0]], three types of stable configurations were discovered (Figure 2a). For example, when the radii of Z-CTTs were low, both the inner and outer tubes broke, with the related configurations belonging to type I (Appendix A). The reason for this is that the slim twin tubes were too close to each other in the initial stage. An extremely high interaction between atoms leads to the breakage of some covalent bonds in the tubes and the simultaneous generation of new bonds between neighboring unsaturated carbon atoms.

When the radii of CTTs became larger, e.g., Z-13, the inner tubes broke seriously and may have connected with the outer tube via sp^3^ bonds (see the inset in the middle column in Figure 2a). This kind of configuration belongs to type II (Appendix A). The VPEA of Z-13 was very different to that of Z-19, due to more bonds being broken in the slim tube.

For type Ⅲ (Appendix A), both the inner and outer tubes were deformed, but displayed no bond breakage in relaxation. For example, Z-20 had a nice stable configuration after relaxation. The value of VPEA with respect to Z-20 jumped down twice due to changes in the shape of the inner tube during relaxation. A detailed discussion will be given below.

When relaxing the armchair CTTs, only type I and type Ⅲ could be found. Similarly, for the slim A-CTTs, their tubes broke seriously due to strong expulsion between tubes. When relaxing thicker CTTs from A-9, no bond breakage occurred during relaxation. When comparing the VPEA curves with respect to A-9 and A-10, we found an obvious difference between the shapes of their inner tubes, e.g., the inner tube in A-9 looked like an “H”, while A-10 had a “V–shaped” inner tube. According to the stable values of VPEA of the two CTTs, we know that the inner tube in A-9 is not well-relaxed due to the smaller decrease of potential energy compared to that in A-10. Theoretically, the configurations provide local buckling solutions [54].

For the type Ⅲ CTTs, both the armchair and zigzag tubes exhibited the same phenomenon, i.e., a thicker CTT displayed a smaller decrease of potential energy per atom, so the system presented less deformation. According to the above discussion, we concluded that Z-CTTs have a stable configuration once the tube is thicker than Z-20, and the A-CTTs have a stable configuration when the tube is thicker than A-9. Both Z-20 and A-9 can be considered minimal-sized CTTs with stable configurations after relaxation at 8 K.

For describing the geometry of the tubes in a stable configuration, both the circumference change rate (C-CR, *β*) and the area change rate (A-CR, *γ*) were calculated and are listed in Table 1. For the A-CTTs, the value of C-CR of the outer tube varied from 110.3% for A-9 to ~95.4% for A-36, i.e., for a slim CTT, its outer tube is under tension after relaxation. A similar conclusion can also be drawn for Z-CTTs. The value of C-CR of the inner tube changed from ~96.3% to 95.4%, i.e., the inner tube shrank with buckling under compression from the outer tube.

After relaxation, the areas of deformed tubes in a CTT were also different to those of the samesingle-walled carbon nanotube, and the values of A-CR of the outer tube decreased from ~117.4% for A-9 to 68.4% for A-36, or from ~126.5% for Z-13 to 71.7% for Z-63. For the inner tubes, their values of A-CR were between 29.3% and 37.0% for either A-CTTs or Z-CTTs. By fitting the values of A-CR and C-CR of the outer tubes (Figure 3), we found that the values obeyed the first-order Gaussian function, with different coefficients for A-CR and C-CR. Meanwhile, for the inner tube, both A-CR and C-CR varied nearby a constant, e.g., 32.5% for A-CR and 95.8% for C-CR. From the fitting results in Figure 3, one can conclude that the A-CR of the outer tubes convergences at 95.3% and the C-CR of the outer tubes tends to be 73.3%.

### 3.2. Comparison of MD and DFT Results

In the above discussion, the VPEA of a CTT was recorded and the final (axial view) configuration of the tube was provided. To show the correctness of the type Ⅲ configurations, here, we adopt DFT to optimize the structure via VASP. By relaxing a DNT, in which the inner and outer shells are bonded, rather than a CTT which has no inter-shell bonds, we can obtain the same result, i.e., a CTT with a concave inner tube. For example, in Figure 4, the DNT ((22,22)) is optimized by DFT (Appendix A). From the first ion step (IS 1) to IS 14, it becomes a CTT with a liable shape. From IS 220 to IS 1120, the shape of the inner tube displays obvious change, which is the same as what happens during the relaxation of the CTT [[22,22]] (i.e., A-22) between 0.6 and 8.4 ps. Later, the inner shell at IS 2000 has a similar shape to that of A-14 at 10.5 ps. According to the VPEA values, the concave shape does not change obviously from IS 1120. The great difference between the final values of VPEA of the two curves produced by MD and DFT simulations is caused by the difference in the bond topology of CTT and DNT. Actually, for the DNT and CTT formed from the same ideal CNTs with large radii, the same stable configuration is formed by DFT-based optimization or MD relaxation.

### 3.3. Geometry and Temperature Effect

Since MD and DFT simulations provided the same results when relaxing/optimizing the CTT and DNT formed from the same ideal CNTs with higher radii, herein, only MD simulation results are provided for revealing the effect of geometry and temperature on the final configurations of the tubes. For simplicity, only three pairs of A-/Z-CTTs are involved in the discussion considering their different radii.

By comparing the first pair of A-14 and Z-25 with radii close to 10 Å, their VPEA curves in Figure 5a indicate that the absolute value of VPEA of the same CTT is higher at a lower temperature, e.g., VPEA = −0.48 eV/atom at 8 K or ~−0.45 eV/atom at 500 K for A-14. Generally, at the same temperature ≤300 K, the absolute value of VPEA of Z-25 is higher than that of A-14. However, at 500 K, the conclusion is not correct. After checking the bond topologies at 500 K shown in Figure 6, we could conclude that bond breakage and generation happen in Z-25 at 500 K. The new bonds within the inner tube do not lead to a greater decrease of VPEA of the system. The reason for this is that the local non-bonding interaction is enhanced due to the existence of new bonds. The new bonds are generated soon after bond breakage in the same tube; however, only parts of the unsaturated atoms are bonded at such a temperature.

In Figure 5b, the VPEA curves of the second pair of CTTs with radii close to 15.5 Å are given. Similarly, at a lower temperature, the VPEA of a CTT has a higher absolute value. At the same temperature, the absolute value of VPEA of the Z-39 is higher than that of the A-22. Their stable configurations are the same (Figure 6). In this case, Z-39 has no new bonds after relaxation.

When the CTTs (e.g., A-36 and Z-63) have radii of ~25 Å or higher, the tubes at 100 K rather than 8 K have the maximal absolute values of VPEA (Figure 5c) within 500 ps of relaxation. The reason for this is that the interaction between the outer and inner tubes at 8 K is stronger than that at 100 K, and the stronger interaction prevents full relaxation of the two tubes, which will have a smaller decrease of potential energy than at 100 K.

An interesting phenomenon, i.e., the two tubes in a CTT exhibit relative sliding, can be discovered when observing the relaxation process. For demonstrating the relative sliding, we labeled a few atoms on both tubes, and the atoms were very close to each other, initially. After 500 ps of relaxation, their distance changed obviously. For example, the labeled atoms on the outer tube (at the tip of the black arrow in Figure 6) and the labeled atoms on the inner tube (in the red circle) may move to opposite sides of the tubes.

In Figure 7, the values of A-CR and C-CR of the CTTs are listed. The effect of temperature on the A-CR of the outer tubes of the CTTs (Figure 7a) is slight, e.g., there is only about a 4% difference of A-CR when the temperature changes about 500 K. For the A-CR of the inner tube, it always varies between 29% and 37%, and the temperature effect can be neglected. For the C-CR of either the inner or outer tube, a temperature increase leads to a difference of no more than 1%. Among the four groups of data in Figure 7, the radius of the ideal CNT, which is used to form the CTTs, has an influence on the A-CR of the outer tube.

### 3.4. Maximal Radii of A-/Z-CTTs with Convex Outer Tubes at 8 K

In Figure 6 or the insets in Figure 2, one can see that the outer tube may have a concave shape after relaxation when the radius of the ideal CNT is high. If the CTTs are made from slim CNTs, their type III stable configurations are convex. Hence, we predict that there is a maximal value of the radius of either A- or Z-CTT which has a convex outer tube after relaxation. To obtain the values, some CTTs were relaxed at 8 K, and the results are listed in Figure 8. To distinguish the shape of the outer tube as being either convex or concave, here, we suggest a mathematical measurement, i.e., for a concave angle *α*, if *α* < 175°, the outer tube is concave. According to this rule, the CTTs A-25 and Z-47 have critical radii, i.e., if an A-CTT has a larger radius than A-25 (with the radius of the ideal CNT being 17.3 Å) or a Z-CTT thicker than Z-47 (with the radius of the ideal CNT being 18.8 Å), it must have a concave outer tube after relaxation at 8 K.

## 4. Conclusions

A concentric twin tube (CTT) can be built by placing a carbon nanotube (CNT) in another identical CNT. Different from diamondene nanotubes, there is no inter-shell covalent bond in a CTT. As a prestressed double-walled nanotube, a CTT is unstable in configuration at a finite temperature. According to molecular dynamics and first-principle calculations, some conclusions are drawn.

First, the tube has three types of configuration after relaxation. For a type I CTT, which is formed from small-radius CNTs, the outer and inner tubes break due to a strong interaction between atoms from the two shells. For a type II CTT, only the inner tube breaks, and may generate a new bond in itself or connecting the outer tube. For a type III CTT, it displays no bond breakage during relaxation, the radii of the two identical CNTs in it are higher, and the inner tube buckles to form a V-shaped cross-section and is confined by the outer tube. The final configuration can also be obtained by optimizing a diamondene nanotube via DFT. Accordingly, we can find the minimal radii of the CTTs of type Ⅲ, e.g., the CTTs from two (20, 0) CNTs or from two (9,9) CNTs at 8 K.

Second, in any one of the type Ⅲ CTTs, supported by a V-shaped inner tube, the outer tube may behave as a convex or concave cross-section. The numerical results indicate that the CTTs from two (25,25) CNTs or two (47,0) CNTs have a critical configuration at 8 K, i.e., for a CTT from two identical CNTs with radius larger than that of (25,25) or (47,0), its outer tube must have a concave cross-section after relaxation.

Third, for a relaxed CTT, the circumferences and areas of the inner and outer tubes are different from those of the corresponding ideal CNT. The area change rate (A-CR) and circumference change rate (C-CR) of a CTT can be calculated. Note that A-CR and C-CR of the outer tube are the first-order Gaussian function of the radius of the ideal CNT, and tend to be 73.3% and 95.3%, respectively. For the inner tube of a CTT, the value of A-CR is between 29.3% and 37.0%, and that of C-CR is close to 95.8%.

Finally, the temperature effect slightly influences the above conclusions.

## Figures and Tables

**Figure 1 nanomaterials-10-00846-f001:**
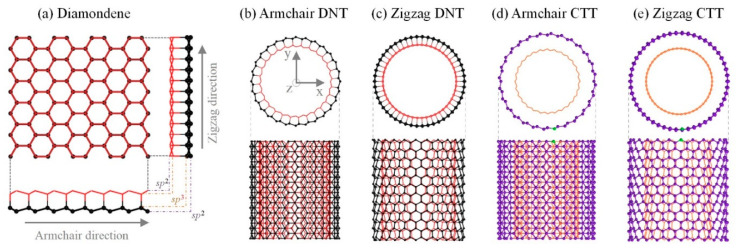
Nanotubes from sp^2^ and/or sp^3^ carbon atoms. (**a**) Diamondene ribbon with both side-views along armchair and zigzag directions. (**b**) Armchair diamondene nanotube (A-DNT) with a chiral index of ((14,14)); (**c**) Zigzag DNT (Z-DNT) with a chiral index of ((25,0)); (**d**) Armchair concentric twin nanotubes (A-CTT) with a chiral index of [[14,14]]; (**e**) Zigzag CTT (Z-CTT) with a chiral index of [[25,0]]. In a DNT, the inner and outer layers are bonded via sp^3^–sp^3^ bonds. In a CTT, the two same tubes are not bonded together.

**Figure 2 nanomaterials-10-00846-f002:**
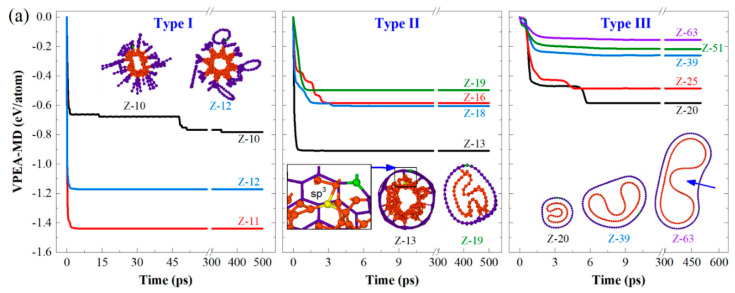
Three types of the final configurations of CTTs after 500 ps of relaxation in an NVT ensemble at 8 K. (**a**) Zigzag CTTs and (**b**) armchair CTTs.

**Figure 3 nanomaterials-10-00846-f003:**
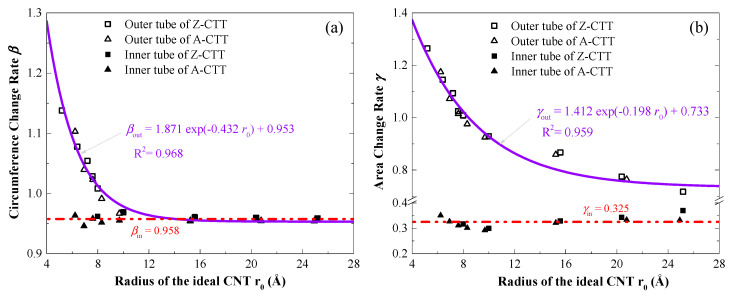
Fitting curves of the circumference change rate (C-CR, *β*) and the area change rate (A-CR, *γ*) of the stable CTTs after relaxation. (**a**) C-CR and (**b**) A-CR.

**Figure 4 nanomaterials-10-00846-f004:**
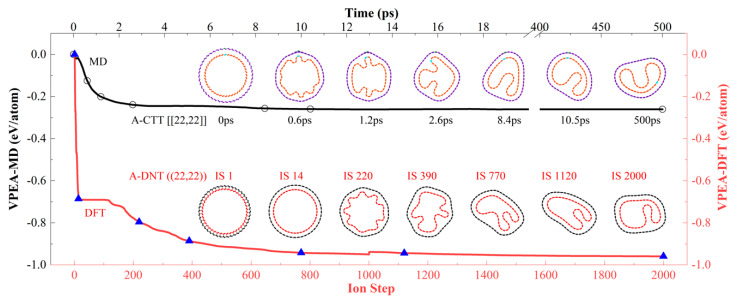
Variation of potential energy per atom (VPEA) and representative snapshots of two types of nanotubes. A-CTT [[22,22]] is relaxed in the NVT ensemble with T = 8 K. For A-DNT ((22,22)), its shape is optimized using a density-functional theory (DFT) calculation within 2000 ion steps (T = 0 K).

**Figure 5 nanomaterials-10-00846-f005:**
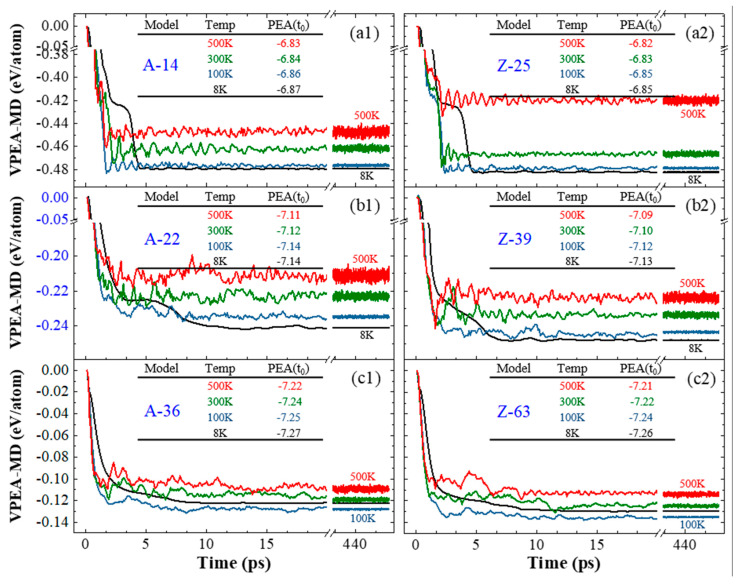
Histories of VPEA of three pairs of A-/Z-CTTs with similar initial radii when relaxed at different temperatures. Radii of ideal CNTs are 9.69 Å for (14,14) (**a1**) versus 9.99 Å for (25,0) (**a2**), 15.23 Å for (22,22) (**b1**) versus 15.59 Å for (39,0) (**b2**), and 24.92 Å for (36,36) (**c1**) versus 25.18 Å for (63,0) (**c2**).

**Figure 6 nanomaterials-10-00846-f006:**
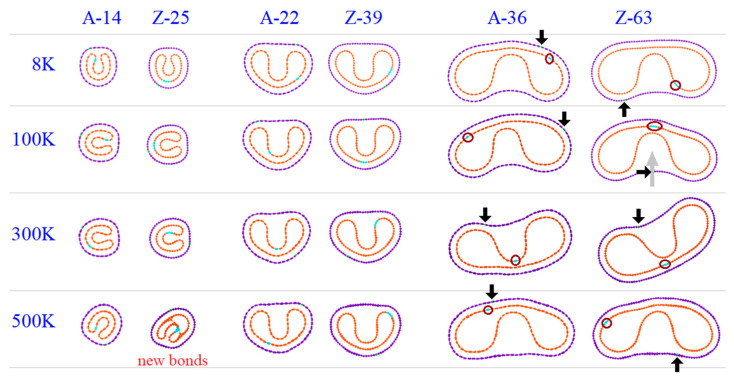
Stable configurations of A-/Z-CTTs with similar initial radii when relaxed at different temperatures.

**Figure 7 nanomaterials-10-00846-f007:**
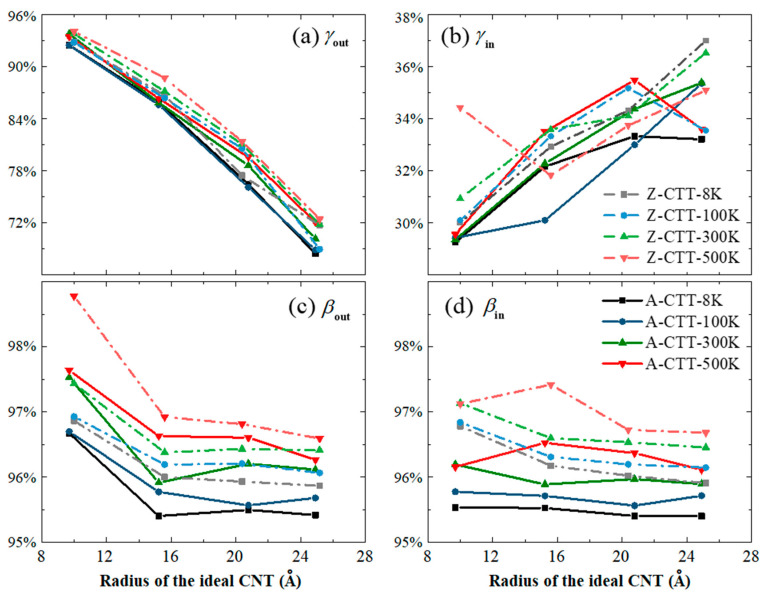
The values of C-CR and A-CR of four pairs of A-/Z-CTTs at different temperatures. (**a**) A-CR *γ*_out_, (**b**) *γ*_in_, (**c**) C-CR *β*_out_, and (**d**) *β*_in_.

**Figure 8 nanomaterials-10-00846-f008:**
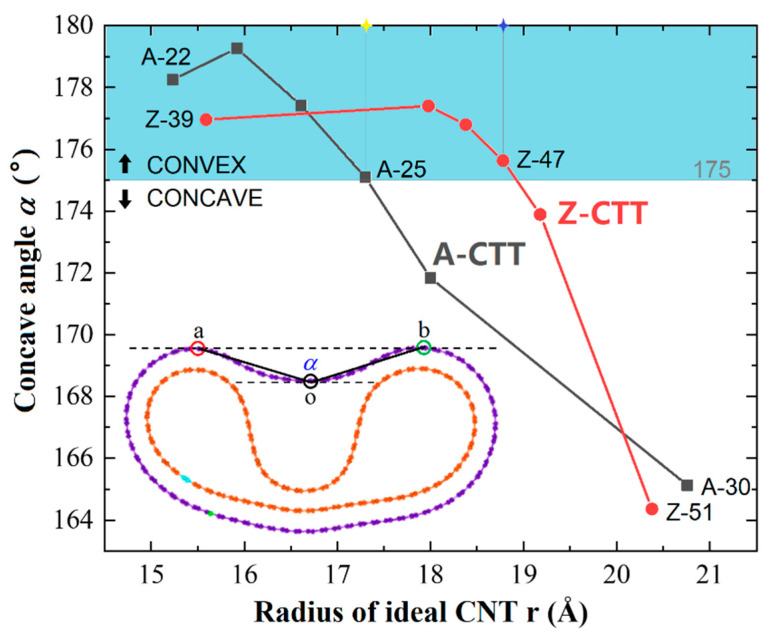
Values of concave angles of the outer tubes in different CTTs at 8 K. Concave angle, i.e., *α*, is the angle between oa and ob, where o, a, and b are the points of tangency between the two parallel dashed lines and the outer tube. Once *α* < 175°, the outer tube is considered as a concave tube.

**Table 1 nanomaterials-10-00846-t001:** Circumference change rate (*β*) and area change rate (*γ*) of the stable CTTs after relaxation. Subscript “0” means the corresponding ideal CNT, and subscripts “in” and “out” mean the inner and outer tubes in a CTT, respectively.

CTT	r_0_ (Å)	A_0_ (Å^2^)	A_out_ (Å^2^)	*γ*_out_ (%)	A_in_ (Å^2^)	*γ*_in_ (%)	C_0_ (Å)	C_out_ (Å)	*β*_out_ (%)	C_in_ (Å)	*β*_in_ (%)
**A-9**	6.23	121.91	143.17	117.43	42.78	35.09	39.14	43.18	**110.31**	37.71	**96.34**
A-10	6.92	150.51	161.37	107.21	49.21	32.70	43.49	45.21	**103.95**	41.14	**94.59**
A-12	8.31	216.74	211.44	97.56	65.38	30.17	52.19	51.73	**99.13**	49.68	**95.19**
A-14	9.69	295.00	272.86	92.49	86.28	29.25	60.89	58.86	**96.67**	58.17	**95.53**
A-22	15.23	728.47	625.74	85.90	234.32	32.17	95.68	91.28	**95.40**	91.40	**95.52**
A-30	20.76	1354.60	1035.73	76.46	451.48	33.33	130.47	124.59	**95.50**	124.47	**95.40**
A-36	24.92	1950.63	1334.81	68.43	647.88	33.21	156.56	149.39	**95.42**	149.36	**95.40**
**Z-13**	5.20	84.80	107.22	126.45	——	——	32.64	37.14	**113.78**	——	**——**
Z-16	6.39	128.45	147.06	114.49	——	——	40.18	43.29	**107.76**	——	**——**
Z-18	7.19	162.57	177.74	109.34	——	——	45.20	47.65	**105.42**	——	**——**
Z-19	7.59	181.13	185.63	102.48	——	——	47.71	49.07	**102.86**	——	**——**
Z-20	7.99	200.70	202.43	100.86	63.64	31.71	50.22	50.63	**100.82**	48.31	**96.20**
Z-25	9.99	313.59	291.29	92.89	94.09	30.00	62.78	60.81	**96.86**	60.75	**96.78**
Z-39	15.59	763.16	661.67	86.70	251.30	32.93	97.93	94.01	**96.00**	94.18	**96.17**
Z-51	20.38	1305.04	1010.96	77.47	447.89	34.32	128.06	122.85	**95.93**	122.97	**96.02**
Z-63	25.18	1991.43	1428.29	71.72	736.97	37.01	158.19	151.65	**95.87**	151.72	**95.91**

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
