# Peer review of "Analogous Diamondene Nanotube Structure Prediction Based on Molecular Dynamics and First-Principle Calculations"

_nanomaterials, 2020, doi:10.3390/nano10050846_

Round 1
Reviewer 1 Report
In this work, the authors propose a new double-walled nanotube formed by two identical CNTs, one inside the other, called concentric twin nanotube (CTT). They apply molecular dynamics and first-principle calculations to investigate the structural stability of CTTs, by considering the radii of the corresponding CNTs, the chiral indexes, and the temperature. They show that at finite temperature, the CTTs have three types of relaxed configurations.
The topic is original and interesting. The paper is rather clear and well-organized. The conclusions are supported by numerical results that, although uncontrollable, seem reasonable.
I support the publication in the journal after a suitable revision.
Here are my suggestions:
- The introduction is rather short and essential. The authors could make a deeper review of the existing literature and better motivate their work. As an example, they mention that “The new nanotubes may have different physical properties to those of CNTs, and they may provide new applications in design of nanodevices based on their unique properties.” They could be more specific and mention some of the key properties and applications they are referring to.
- I believe that the authors have used some automatic tool to reference the figures in the paper. However, in the pdf on my computer, all figures are pointed with "0". The authors should consider this potential issue.
- The CTT can be built by putting a carbon nanotube (CNT) into another identical CNT. Has this been experimentally tried in any way? Did the author check for experimental evidence to support any of their conclusions about the three types of configurations that they have identified based on their calculations?
- Besides the fundamental interest, is there any practical relevance for this study? Can the authors envisage any application of the CCTs, for instance of the type III ones?
Reviewer 2 Report
The manuscript by Zhou et al. presents the model of a hypothetical allotrope of carbon. It can be derived from two carbon nanotubes of the same size, yet, with the different cross-sections: internal one is collapsed, while external one adapts oval-like shape. The authors have studied profoundly morphology, stability and thermal stability of both zigzag and armchair nanotubesusing two levels of theory - AIREBO force-field in MD regime and DFT method as the independent and trustworthy cross-check. Honestly, I did not find a lack from scientific point-of-view. These nanostructures are quite hypothetical and, strangely, have been not observed experimentally, yet. Though, the subject of this study belongs to the topics of Nanomaterials. I recommend this manuscript for publication in Nanomaterials after minor revision of mostly representative character.
1) English grammar is fine, but the style of expressions requires an improvement. I would give only two examples from first page like "Carbon materials in nature or man-made are rich...", "...people predicted that..." etc.
2) The notation of twin nanotubes in the text and in the figure captions is not fully clear: what is the difference between notations using round and square brackets like [[12,12]] and ((12,12))? The classification must be single.
3) The links on figures and on tables throughout the text are wrong (they all refer to 0).
4) Methodology. The authors wrote: "The models are put in a periodic box with square cross-section, and the side length is 15.45 Å..." Obviously, there is a mistake. The radii of several nanotubes given in table 1 is larger, than this value.
5) Methodology. The authors should give in this section the temperatures used for MD NVT. I wonder, why the authors have used mostly T= 8 K? Why not 300 K?
6) Supporting materials, captions of movies. What does notations like "[0.1, 5.0] ps" and "[0,1200]" mean? Once again, both round and square brackets are used for chirality description.
Reviewer 3 Report
This is a very nice contribution in study of nanomaterials. The autors used state of art methods using MD and ab initio techniques to describe the new type of nanotubes. The results are cleary presented and convincing. I recommend to publish the paper as it stands.
Author Response
Thank you for your understanding of our work.
Reviewer 4 Report
The authors present a combined MD and DFT investigation of the stability of twinned CNTs in dependence of the tube diameter. The results are interesting and new, but it is not clear to me, whether the results are biased by the initial conditions. Therefore I have a few essential remarks the authors should consider before publication. Minor remarks I marked in the attached pdf-file of the manuscript.
1. For the starting structure only the distance between the two SWCNT is given, but it not clear to me, to what extend the inner CNT is compressed and the outer is extended. It would be for the whole manuscript much better to discuss the effect on the C-C distance in the tubes. Because the interactomic potential is anharmonic around the equilibrium, a compression with the same distance cost much more energy than the elongation.
2. It is not clearly described how the DFT "optimization is performed". It seems that it is a structure optimization (at least the 0K looks like this), but then I am surprised, that 2000 steps are necessary until convergence.
3. It seems that for twinned CNT with larger diameter always the same structural motives occur, also there it would be very interesting to discuss the C-C distance in the inner and outer CNT, not only the averages diameter and area.
4. Has there been a study by the authors on the dependence of the starting distance of the twinned CNT. Or in other words. How stable is the found structure with regard to the starting distance.
5. Although the reference to the figures and tables is linked in the pdf, for publishing I would prefer the old-fashioning style with written out references, but that is more a remark to the editor.
In summary, although it is probably quite difficult to make such structures experimentally, it is an interesting computational study, if the new found structural motifs are stable with respect to the starting geometry.

Round 2
Reviewer 1 Report
The authors have considered all my comments and suggestions. They have made the essential corrections to the manuscript and provided satisfying replies. The paper can be accepted as is.
Reviewer 4 Report
The authors have argued in the response and some parts are now clearer to me. But still it is my opinion, that the scientific content can be improved a lot, if they would discuss the C-C bond length issue. I understand that that is a lot of work, because an automatic algorithm has to be programmed to analyse the bond distances. There exist such analysis tools.
But I understand the authors, that they want to publish the article as it is. In the present form I think it does not deserve the highest rates, but it presents an interesting study and one can built on the results. Therefore I agree that it can be published as it is.